# Machine vision model for detection of foreign substances at the bottom of empty large volume parenteral

**Pi Yuan, Chen Li©\*, Peng Tang, Bin Yuan, Yongjing Yin**

School of Mechanical and Energy Engineering, Zhejiang University of Science and Technology, Hangzhou, China

\* lee_lichen@163.com

**Data Availability Statement:** All relevant data are within the manuscript and its Supporting Information files.

**Funding:** This work was supported by Zhejiang Provincial Natural Science Foundation of China

## Abstract

Empty large volume parenteral (LVP) bottle has irregular shape and narrow opening, and its detection accuracy of the foreign substances at the bottom is higher than that of ordinary packaging bottles. The current traditional detection method for the bottom of LVP bottles is to directly use manual visual inspection, which involves high labor intensity and is prone to visual fatigue and quality fluctuations, resulting in limited applicability for the detection of the bottom of LVP bottles. A geometric constraint-based detection model (GCBDM) has been proposed, which combines the imaging model and the shape characteristics of the bottle to construct a constraint model of the imaging parameters, according to the detection accuracy and the field of view. Then, the imaging model is designed and optimized for the detection. Further, the generalized GCBDM has been adopted to different bottle bottom detection scenarios, such as cough syrup and capsule medicine bottles by changing the target parameters of the model. The GCBDM, on the one hand, can avoid the information at the bottom being blocked by the narrow opening in the imaging optical path. On the other hand, by calculating the maximum position deviation between the center of visual inspection and the center of the bottom, it can provide the basis for the accuracy design of the transmission mechanism in the inspection, thus further ensuring the stability of the detection.

## 1. Introduction

Large Volume Parenteral (LVP) is one of the five important classes of pharmaceutical products. Its safety is closely related to the life and health of patients, and the quality detection of LVP needs strict requirements [1]. Due to the constraints of manufacturing technique and environment, in some cases, there may be some foreign substances inside the LVP fluid, such as glass chips, fibers, hairs, plastic chips, etc. When the liquid medicines containing foreign substances have been taken into the human body, the patients may catch a fever or chill, and in some case, even face death [2]. Therefore, it is necessary to conduct vial-by-vial detection for insoluble foreign substances in LVP.

For the detection of foreign substances in LVP, the traditional detection method is directly through the manual visual, which is labor-intensive and prone to visual fatigue and production

under Grant No. LGF20F050002, National Natural Science Foundation of China under Grant No. 62103340, and Hangzhou City Agriculture and Social Development General Project under Grant No. 20191203B34 and No.20201203B118. The fund supported the corresponding author Chen Li. Author Chen Li participated in the writing of the paper and the verification and guidance of the experiments.

**Competing interests:** The authors have declared that no competing interests exist.

quality fluctuations. With the development of computer vision technology, some machine vision inspection equipment has been applied to the detection of foreign objects in pharmaceutical fluids [3, 4]. In general, these devices often rotate the liquid bottle by mechanical means, thus rotating the foreign objects to follow the movement of the liquid. Then the rotating mechanism is stopped sharply and the motion of the foreign objects are observed by continuous photography of the vision system [5, 6], and finally the identification of different targets is performed [7]. At present, the detection technology for foreign substances in pharmaceutical solutions is more mature [8], but most of the above detection equipment is concentrated in the detection after the filling of pharmaceutical solutions. However, the plastic empty bottles used for pharmaceutical liquid packaging may have foreign substances brought in and attached to the bottom or body of the bottle before filling the liquid. After passing through the subsequent filling process, they become substandard products. In order to reduce production costs, the empty plastic bottles after the blow molding process and before filling also need to be detected for foreign substances.

For the above-mentioned foreign substances adsorbed on the side of the LVP bottle, drawing on the traditional detection in transparent glass bottles [9–11], the imaging and detection of foreign substances can be quickly achieved by backward illumination of a planar light source. However, in the detection of foreign substances adsorbed at the bottom of LVP bottle, due to the special characteristics of the bottle (as shown in Fig 1), there will be differences in imaging compared to the traditional transparent glass bottles and will interfere with the illumination or imaging optical path [12]. For example, the presence of a special structure like a pull ring at the bottom can obscure the imaging at the bottom at some imaging angles. In addition, the narrow bottle opening will interfere with the imaging light path at the bottom.

With reference to the traditional detection of defects in glass bottles, such as the detection of empty beer bottles, empty beverage bottles, etc. [13–20], the following methods are commonly used to detect the bottom of bottles:

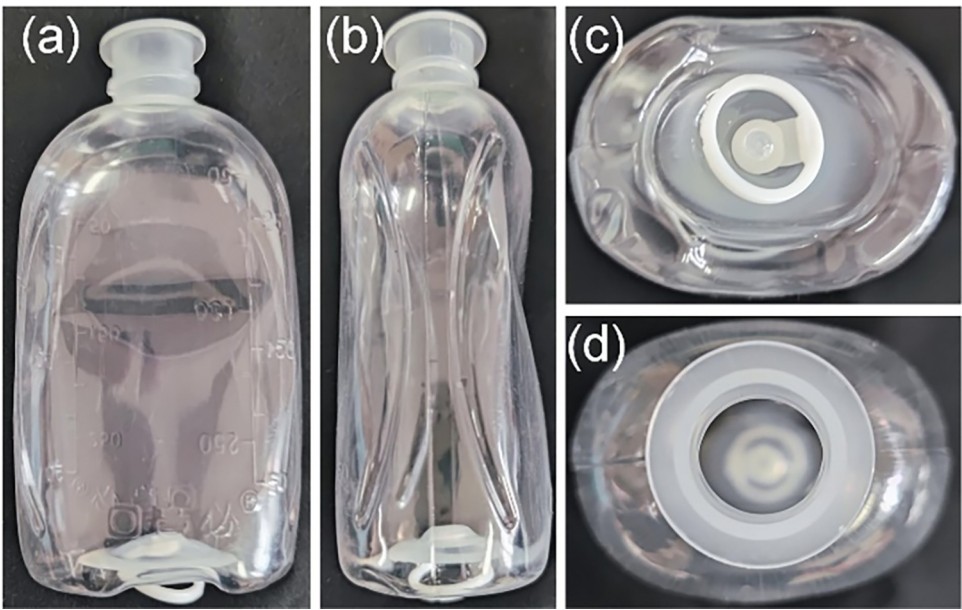

**Fig 1. Images of commonly used LVP bottle.** (a) front view; (b) side view; (c) image of the bottom; (d) image of the opening.

1. Illumination from the bottom and imaging through the opening of the bottle [21–25], as shown in Fig 2(A) and 2(B). The detection model shown in (a) uses a uniform planar light source to illuminate the bottom and the imaging unit to image the bottom through the opening. The detection model shown in (b) uses ring light at a specific angle to illuminate the bottom uniformly according to the specific shape of the bottom, and to avoid the interference of the raised pattern of bottom [26, 27]. Similarly, the imaging unit images the bottom through the mouth of the bottle. The two methods mentioned above, which illuminate the bottom uniformly [18], simply capture an image of the bottom. But they don't take into account constraints such as detection accuracy and the shape of the bottle opening. These methods can achieve uniformed image for bottle bottom inspection with low detection accuracy, or with a small detection field of view. Such as the traditional glass beer bottles and so on [28], the detection accuracy of which is about $3mm \times 3mm$. In addition, for the bottom of the LVP bottle (Fig 1(C)), the pull ring at the bottom is not at a fixed angle as well as a fixed height, and it is not possible to use the ring light to illuminate the bottom of the bottle evenly.

2. Coaxial illumination and imaging through the bottle opening [29], as shown in Fig 2(C). The coaxial light source incident a uniform plane light through the beam splitter to the bottom, and the imaging unit images the bottom easily through the opening of the bottle. This method is mainly applicable to the inspection of bottles with wide opening, such as canning jars [30, 31]. For the bottom detection of LVP, the small opening (as shown in Fig 1(A) and 1(D)) will interfere with the incident light, resulting in poor illumination uniformity at the bottom and subsequent inability to obtain clear image.

3. Direct illumination and direct imaging the bottom, as shown in Fig 2(D). Specific angle ring light incident on the bottom, forming bright-field or dark-field illumination. And the imaging unit directly imaging the bottom, and obtains the dark-field or bright-field image of the bottom. This detection method is mainly applicable to the bottom detection of the more regular glass bottle, such as beer bottle or glass LVP bottle bottom. For plastic LVP bottle bottom detection, the presence of pull ring and irregular shapes formed after the blow molding, can prevent the formation of bright-field or dark-field in the imaging, thus failing to obtain clear images.

4. Illumination through the opening of the bottle and direct imaging the bottom, as shown in Fig 2(E). The illumination of the bottom is completed by a uniform planar light source

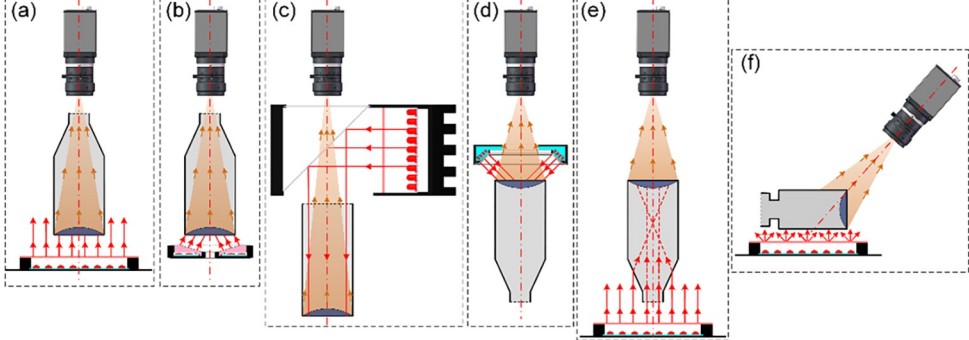

**Fig 2. Imaging solutions for machine vision in traditional bottle bottoms.** (a) (b) illumination from the bottom and imaging through the opening of the bottle; (c) coaxial illumination and imaging through the bottle opening; (d) direct illumination and direct imaging the bottom; (e) illumination through the opening of the bottle and direct imaging the bottom; (f) illumination from the side of the bottle and tilt imaging the bottom.

incident from the opening, and the imaging unit images the bottom directly from the back. However, for LVP with narrow bottle opening, this detection method will form a bright spot at the opening position, which will easily interfere with the imaging of foreign substances to be detected.

5. llumination from the side of the bottle and tilt imaging the bottom, as shown in Fig 2(F). Uniform plane light source evenly illuminates the side of the bottle, and the imaging unit is tilted at a certain angle to achieve the detection imaging of the bottom. This approach is more conducive to the integration of inspection units, i.e., the simultaneous realization of the inspection of the side of the bottle and the inspection of the bottom, and is commonly used for the rapid inspection of small glass (cylindroid) bottles. However, this detection method is susceptible to interference from the side imaging of the bottle, and the tilt imaging of the bottom is likely to cause the image out of focus.

In summary, the current detection of the bottom of plastic LVP bottle is mainly due to the irregular shape and the narrow opening of the bottle, which will cause interference with the incident light and the imaging of the bottom in the inspection. In addition, its detection accuracy is generally better than 50$um$, which is much higher than the detection accuracy of traditional glass bottles. The existing inspection methods do not have a universal visual inspection model based on the inspection accuracy, the shape of the bottle, and the field of view of the inspection.

In this thesis, a machine vision model and system for bottom detection in empty LVP bottle, namely geometric constraint-based detection model GCBDM, is developed to overcome the shortcomings of existing detection techniques. It completes the design and optimization of the imaging model of the bottom according to the requirements of inspection accuracy and field of view, combined with the imaging model and the shape characteristics of the bottle to be inspected. Further, by modifying the relevant constraint parameters in different application scenarios, it is possible to adapt to different detection needs. The innovative application of this paper consists of the following main aspects:

1. Taking empty LVP bottle as the research object, this paper establishes a generalized visual detection model for the bottom detection based on the imaging theory and the constraints in the actual scenario. The parameters of the vision inspection hardware are optimized according to the geometry of the bottle and the inspection accuracy, combined with the imaging model, etc., so that a more economical vision inspection hardware solution can be designed. It is also possible to image the bottom completely on the camera imaging plane through the narrow opening/neck of the bottle.

2. The GCBDM can solve the problem of generality of bottle bottom detection in different detection scenarios, taking different pharmaceutical packaging products as examples, such as bottom detection for cough syrup bottle and capsule medicine bottle with different detection needs. And, by calculating the maximum position deviation range between the center of visual inspection and the center of the target to be inspected, it can provide the basis for the accuracy design of the transmission mechanism in the inspection.

The remainder of this paper is organized as follows. Section 2 focuses on the bottom inspection of empty LVP bottle as the research object, introducing the geometric characteristics of the sample to be inspected, the inspection accuracy, the constraints, and further building the applicable machine vision inspection system. Section 3 builds the geometric constraint- based detection model (GCBDM) based on the detection accuracy and detection constraints in

Section 2. The vision model can achieve clear imaging of the bottom under the condition of ensuring the inspection accuracy and inspection field of view, and it can avoid the interference of special form factor such as narrow bottle opening during the imaging process. Section 4 takes empty LVP bottle as an example, designs a machine vision imaging system according to its inspection requirements, and calculates a variety of hardware combination solutions, and gets the optimal machine vision solution by comparing the experimental results. In addition, taking the bottom detection of other commonly used bottles in the pharmaceutical packaging field as an example, such as the bottom detection of cough syrup bottle and capsule medicine bottle, the optimization of the visual detection system is completed by modifying the constraints according to the GCBDM established in Section 3, and the generality of the model is verified by the experimental results. Finally, a brief conclusion is presented in Section 5.

## 2. Machine vision system for bottle bottom inspection

This paper takes the bottom detection of empty LVP bottle (as shown in Fig 1) as the object of study. Compared with bottoms detection of traditional bottles, such as beer bottles and can bottles, the detection accuracy detection of LVP bottle is about $0.05mm \times 0.05mm$, while the detection accuracy of the latter is only $3mm \times 3mm$ but the two have similar detection field of view, thus the former has a higher detection difficulty. In addition, the former has a narrow bottle opening, which tends to interfere with the illumination of the bottom as well as the imaging optical path. Therefore, the design and optimization of the whole machine vision system needs to be completed according to the geometry constraints of the sample to be inspected, the constraints of the inspection field of view, and the constraints of the inspection accuracy.

Firstly, considering the special geometry of LVP bottle, is not the traditional cylindrical. For the imaging unit, this paper adopts the pin-hole imaging model with the effect of near large and far small, which ensures that the near imaging height can pass through the narrow bottle opening. That is, imaging from the bottom of the bottle can pass through the narrow mouth of the bottle. For the light unit, it is difficult to meet the ideal inspection conditions such as bright field and dark field using ring light incidence due to the presence of a pull ring and the uneven surface at the bottom. In addition, due to the narrow opening of the bottle, it is also difficult for the coaxial light to be incident from the opening to the bottom of the bottle, thus making it difficult to achieve uniform illumination of the bottom. At the same time, because the bottle is not cylindrical, it is difficult to achieve a uniform image from the side incident light and from the side tilt imaging. And its bottle bottom size is large, tilt imaging is more likely to produce out-of-focus, resulting in unclear imaging. Based on the above analysis, in order to achieve uniform illumination of the bottom, this paper uses gross glass and LED plane light to form uniform illumination, which is incident from the lower part to the bottom. As shown in Fig 3.

In the inspection system shown in Fig 3, the drive mechanism moves in direction, using mechanical devices such as cylinders and clamps to hold the neck of the bottle and drive the bottom within the inspection range. The light under the bottom is evenly incident on the bottom, and the Factory Automation (FA) lens is used to capture the information from the bottom through the narrow opening into the image sensor, using the pin-hole imaging theory. According to the optical path shown in Fig 3, in order to ensure the effectiveness of the imaging optical path, then the following conditions need to be met:

1. the resolution of the imaging should be smaller than the minimum size of the detection target, ensuring clear imaging of the detection target;

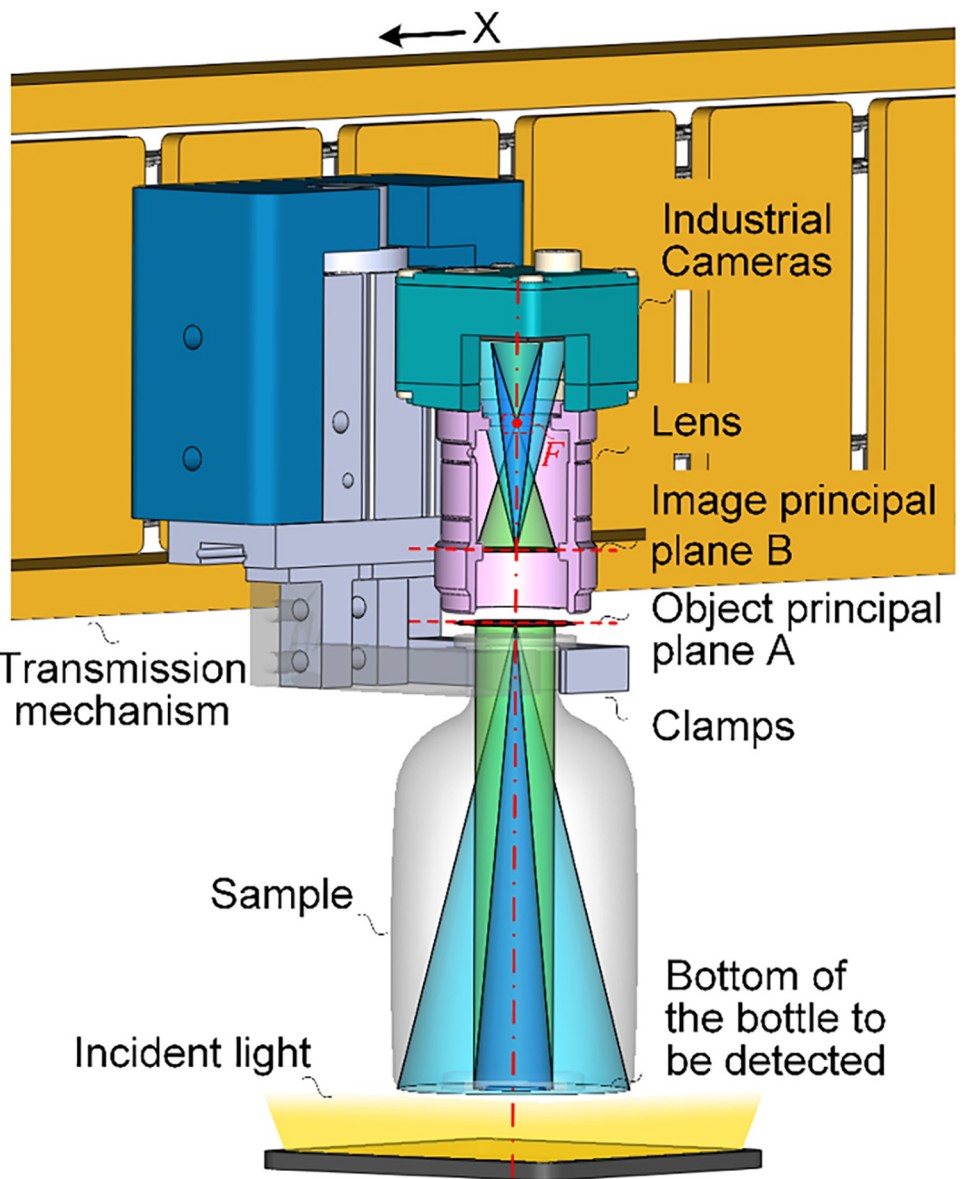

**Fig 3. System structure diagram of LVP bottle bottom detection.**

2. the detection field of view corresponding to the shortest side of the camera sensor needs to be larger than the longest dimension of the bottom of the bottle being inspected, which can ensure that the detection target is completely presented within the camera's field of view;

3. the working distance of the imaging unit needs to be greater than the height of the bottle, which can avoid the occurrence of defocusing and resulting in unclear imaging;

4. to avoid interference from the narrow bottle mouth on the imaging of the bottle bottom, it is necessary to ensure that the imaging parameters at the bottle mouth position are smaller than the diameter of the bottle mouth, which can effectively avoid occlusion of the imaging optical path by the bottle mouth.

## 3. Geometric Constraint-Based Detection Model (GCBDM)

The system diagram shown in Fig 3 is simplified to the model schematic shown in Fig 4(A). Uniform plane illumination, through the bottom into the imaging system, the bottom information ($x$) imaged on the camera imaging plane ($x'$). Since the bottom is approximately elliptical, the long axis of the elliptical bottom is imaged on the short side of the camera imaging plane in order to ensure the integrity of the bottom imaging. Where the length of the long axis is $D_1$. For the camera and the lens composed of the pin-hole imaging unit, the focal point is $F$, the focal distance is $f$, the focus-image distance is $i$, the object principal plane of the imaging is $A$, and the image principal plane is $B$. Based on the analysis of Section 2, in order to ensure that the imaging system can achieve complete imaging of the bottom through the width of the bottle opening ($w$), four different sets of constraints on the imaging system are required from the inspection requirements, as well as the geometric parameters of the bottle. The calculation flow of the constraints and the parameters of the imaging system are shown in Fig 4(B).

The flow chart shown in Fig 4(B) focuses on the calculation of the dimension ($P$) and pixel size ($p$) of the camera, as well as the focal length ($f$) and magnification ($M$) of the lens, based on different constraints. This facilitates the design of the optimal imaging unit configuration.

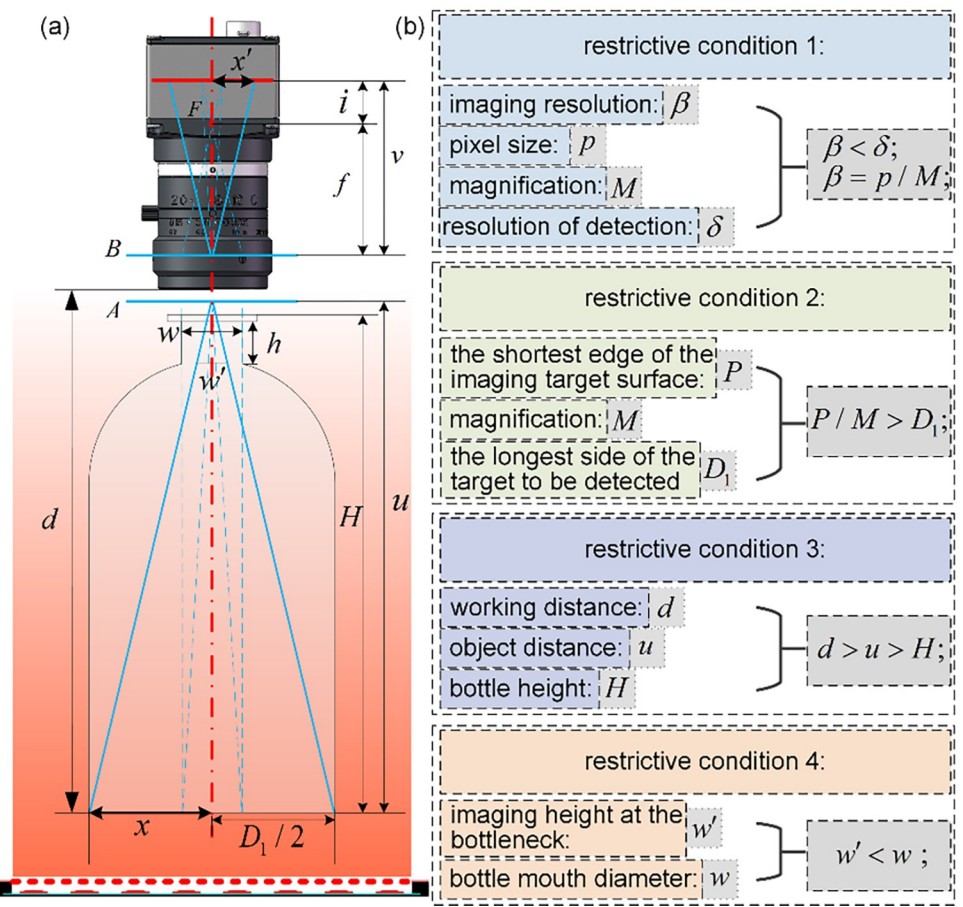

**Fig 4. Schematic diagram of GCBDM.** (a) schematic diagram of bottle bottom detection; (b) flow chart of imaging parameters calculation based on different constraints.

First, according to the principle of lens imaging:

$$\frac{1}{f} = \frac{1}{u} + \frac{1}{v} \tag{1}$$

$$v = i + f \tag{2}$$

In Fig 4, using the principle of similar triangles:

$$\frac{i}{f} = \frac{x'}{x} = M \tag{3}$$

Where $x$ is the object height of the image, $x'$ is the image height of the image, and $M$ is the magnification of the image.

From Eqs (1)(2)(3), it can be found that:

$$u = \frac{1+M}{M} \cdot f \tag{4}$$

According to constraint 1: the resolution of the imaging ($\beta$) should be smaller than the minimum size of the target to be detected ($\delta$):

$$\frac{p}{M} = \beta < \delta \tag{5}$$

According to constraint 2: the inspection field of view ($2x$) is larger than the maximum size of the bottom to be inspected ($D_1$):

$$\frac{P}{M} = 2x > D_1 \tag{6}$$

It should be noted that $P$ is the short side dimension of the camera sensor size, and its dimension is the product of the number of pixels $Pixel_{short}$ on the short side of the sensor and the pixel size $p$. In addition, the bottom of the LVP bottle is approximated as an ellipse, and the long axis of the ellipse is $D_2$, and the short axis is $D_1$, and the short axis is $D_2$.

According to Eqs (5)(6):

$$\frac{p}{\delta} < M < \frac{P}{D_1} \tag{7}$$

According to Formula 7: A balance point can be found that ensures sufficient resolution to detect the required minimum features, while also ensuring that the object to be inspected can be completely captured within the camera's field of view. If the magnification is too high, it may lead to a too small field of view, unable to fully observe the entire object to be inspected; if the magnification is too low, it may be unable to discern the fine features that need to be detected.

According to constraint 3: the working distance of the imaging unit $d$ needs to be greater than the height of the bottle $H$. In order to further ensure the safe distance between the lens and the bottle, the distance between the object principal plane of the imaging $A$ and the bottom (the object distance $u$) is approximated as the safe distance for imaging:

$$d > u > H \tag{8}$$

According to constraint 4: the width of the imaging at the opening of the bottle $w'$ is smaller than the diameter of the opening $w$:

$$w' < w \tag{9}$$

Where, according to the principle of similar triangles, the bottom $D_1$ passes through the opening of the bottle, the corresponding imaging width $w'$ is calculated as follows:

$$\frac{u - (H - h)}{u} = \frac{w'}{D_1} \tag{10}$$

$$w' = D_1 \cdot \frac{u - (H - h)}{u} \tag{11}$$

Where, $h$ is the height of the bottleneck.
According to Eqs (8)(9)(11), it can be obtained that:

$$H < u < \frac{D_1 \cdot (H - h)}{D_1 - w} \tag{12}$$

Further, according to Eqs (4)(12):

$$H < f \cdot \left(1 + \frac{1}{M}\right) < \frac{D_1 \cdot (H - h)}{D_1 - w} \tag{13}$$

$$\frac{1}{\frac{D_1 \cdot (H-h)}{f \cdot (D_1 - w)} - 1} < M < \frac{1}{\frac{H}{f} - 1} \tag{14}$$

In industrial cameras and lenses for vision inspection, the camera dimension $P$ the pixel size $P$ and lens focal length $f$ are a series of fixed parameters. And, for the shape parameter ($H$, $h$,$D_1$) of the sample to be tested and the minimum size of the target to be detected ($\delta$)are the known values. Therefore, the imaging system parameters matching the inspection requirements can be calculated according to Eqs (7)(14), which can avoid interfering factors in imaging and ensure the integrity and stability of imaging, i.e., the information of the bottom of the bottle in imaging is not obscured by the narrow opening of the bottle.

In summary, by designing the GCBDM model and selecting an appropriate hardware solution, clear imaging of the bottom of the bottle can be achieved. Moreover, after obtaining a clear image of the bottom of the bottle using the imaging method designed in this paper, image processing of defects can be easily operated. For example, some methods of feature enhancement, and Blob analysis can be easily used in the detection of foreign objects at the bottom of the bottle, which are shown as follows:

(1) Feature enhancement.

Feature enhancement can increase the contrast information of the object to be recognized in the background, ensuring a clear distinction between foreign objects and the clean bottom area of the bottle. Its principle is shown in Eq (15):

$$I_{em} = \text{round}((I - mean_{m \times m}) * K_{em}) + I \tag{15}$$

Where,$I_{em}$ is the enhanced image, $mean_{m \times m}$ is the mean value in the image $I$ within the window size $m \times m$, and $K_{em}$ is the enhancement factor ($K_{em} > 1$).Through the above image

processing operations, foreign objects at the bottom of the bottle can be clearly identified, resulting in an image with strong contrast.

(2) Blob Analysis.

Formula (15) can further enhance the background and contrast of the image. The enhanced image can then be used for defect extraction and detection through an adaptive global threshold segmentation algorithm. Further analysis of blob features can be performed to label the defects, based on the principles presented in Eqs (16) and (17).

$$I_{bottle} = \text{OTSU}(I_{em}) \tag{16}$$

$$I_{result} = \text{Blob}(I_{bottle}) \tag{17}$$

In conclusion, with the use of GCBDM in this paper, a clear image of the bottom of the bottle can be achieved. Then, the information of defects can be extracted easily using the traditional image processing of defect detection.

# 4. Experiments

## 4.1 Experiments of GCBDM on the Bottle Bottom of LVP

**4.1.1 Constraints of detection, and calculation of GCBDM.** In this paper, the object of study is an empty LVP bottle whose dimensions are shown in Fig 4(A), and whose bottom is an approximate ellipse with the long and short axes respectively $D_1 = 70mm$, $D_2 = 50mm$ In addition, the required geometry parameters in the calculation process are $w = 18mm$, $h = 18mm$, $H = 142mm$ In addition, the detection accuracy of foreign objects in LVP is $\delta = 00.5mm$. Through the above-mentioned shape characteristics of empty LVP bottle and the inspection requirements, the calculation of vision parameters is completed. And according to the commonly used industrial camera and industrial lens parameters, design a suitable imaging configuration, and complete the imaging of the bottom. Therefore, based on the given parameters of $\beta < \delta$, $\beta = p/M$, $P/M > D_1$, $d > u > H$, and $w' < w$, as well as the four constraint conditions, the computational process is obtained as shown in Fig 5.

According to the above calculation process, the main parameters of the camera unit in the detection model are the resolution of the camera, the sensor size, and the pixel size. Among them, parameters of the commonly used industrial camera and lens are shown in Table 1.

It should be noted that, considering that the bottom may have the problem of rotation, that is, the shortest edge of the field of view is required to meet the requirements of greater than $D_1$ = $70mm$, and the detection accuracy is better than $03 \delta = 0.05mm$. So, for the industrial

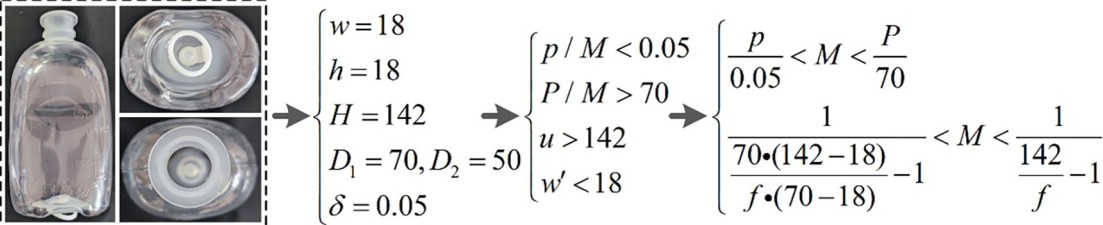

**Fig 5. Constraints of LVP bottle bottom detection and calculation process of detection model.**

**Table 1. Parameters of common industrial cameras and lenses.**

| | Resolution | Sensor size | Pixel size | Lens focal length |
|---|---|---|---|---|
| Parameters 1 (1.3MP Area Scan Camera) | 1280×1024 | 1/2″ | 4.8um×4.8um | Compatible with the sensor size: $f = 6mm/8mm/12mm/16mm$ |
| Parameters 2 (2.3MP Area Scan Camera) | 1920×1200 | 2/3″ | 4.8um×4.8um | |
| Parameters 3 (3.2MP Area Scan Camera) | 2048×1536 | 1/18″ | 3.45um×3.45um | |
| Parameters 4 (5MP Area Scan Camera) | 2448×2048 | 2/3″ | 3.45um×3.45um | |
| Parameters 5 (20MP Area Scan Camera) | 5472×3648 | 1′ | 2.4um×2.4um | |

camera, the number of pixels on the short side of the sensor size should meet:

$$Pixel_{short} > \frac{D_1}{\delta} = 1400 \tag{18}$$

Therefore, based on the requirements in Eq (18) and the parameters of industrial cameras and lens, the parameters that may be suitable for the imaging are designed. The magnification $M$ is calculated for different hardware conditions using Eqs (7)(14), and then can be used as a criterion for camera and lens matching. That is, the $M$ calculated by the camera parameters and the $M$ calculated by the lens parameters have the same interval range, shown in Table 2.

Based on the above calculations and configurations, some economical imaging solutions are designed in terms of actual costs as shown in Table 3.

**4.1.2 Experiment results and discussion.** According to the schemes in Table 3, the information at the bottom of the LVP bottle, through the narrow bottle opening, is imaged on the camera imaging plane, and the experimental results are shown in Fig 6.

As can be seen from the Fig 6, all four imaging schemes can achieve complete imaging of the bottom and the foreign object. And all of them can meet the resolution. However, for the imaging methods with large sensor size (20MP) in (b)(c)(d), the bottom information occupies only a small part of the image, i.e., there is more waste in the camera imaging plane. Therefore, the optimal imaging method is Option 1.

In addition, it should be noted that in the above-mentioned detection, the bottle bottom in the transmission process, there will be a certain error in the position of each photo. That is, the photo optical axis and the central axis of the bottom will not be completely coincident. If the deviation between the two central axes is large, i.e., the width of the imaging model at the bottle neck position may exceed the width of the bottle opening. That is, the maximum deviation range of the imaging center from the center of the bottom during the inspection. And, in the

**Table 2. Magnification calculated based on constraints for different hardware conditions in the bottom detection of empty LVP bottle.**

| | | $M$ |
|---|---|---|
| $f$ | 6mm | 0.0373<$M$<0.0441 |
| | 8mm | 0.0504<$M$<0.0597 |
| | 12mm | 0.0775<$M$<0.0923 |
| | 16mm | 0.1061<$M$<0.1270 |
| $Pixel_{short}$ | 1536 | 0.0.670<$M$<0.0757 |
| $P$ | 3.45um | |
| $Pixel_{short}$ | 2048 | 0.0690<$M$<0.1009 |
| $P$ | 3.45um | |
| $Pixel_{short}$ | 3648 | 0.0480<$M$<0.1251 |
| $P$ | 2.4um | |

**Table 3. Feasible imaging solutions for the bottom detection of empty LVP bottle.**

| Option 1 | Camera: 2448×2048, 3.45um×3.45um Lens:$f = 12mm$ support sensor size 2/3″ | Approximate working distance: $f = 12mm$: $142.011mm < u < 166.83mm$ | | $\Delta w = w - w'$ $f = 12mm$: $0.026mm < \Delta w < 9.122mm$ | |
|---|---|---|---|---|---|
| Option 2 | Camera: 5472×3648, 24um×24um Lens:$f = 8mm/12mm/16mm$ support sensor size 1″ | Approximate working distance: | | $\Delta w = w - w'$ | |
| | | $f = 8mm$ : | $142.003mm < u < 166.730mm$ | $f = 8mm$ : | $0.060mm < \Delta w < 9.1256mm$ |
| | | $f = 12mm$ : | $142.011mm < u < 166.639mm$ | $f = 12mm$ : | $0.088mm < \Delta w < 9.122mm$ |
| | | $f = 16mm$ : | $143.898mm < u < 166.801mm$ | $f = 16mm$ : | $0.038mm < \Delta w < 8.320mm$ |

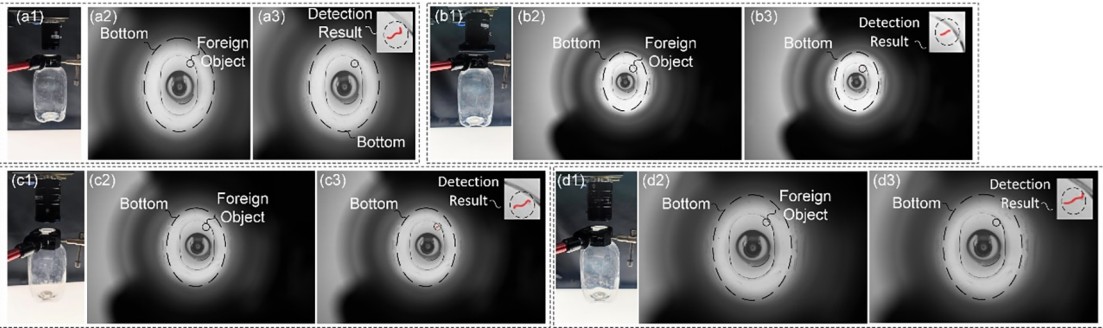

**Fig 6. Imaging results obtained according to different detection schemes.** (a1) (a2) the constructed detection model and the image of the bottom acquired by the 5MP camera at the lens focal length f = 12mm; (b1) (b2) (c1) (c2) (d1) (d2) the constructed detection models and the images of the bottom acquired by the 20MP camera at the lens focal length f = 8mm,12mm,16mm; (a3) (b3) (c3) (d3) the detection results of the traditional image processing with the Eqs (15) (16) and (17).

above calculation, the maximum deviation value is larger when the working distance is closer, thus reducing the requirement for mechanism drive accuracy. Therefore, under the condition that the mechanical installation height allows, the detection height is appropriately reduced. In addition, in the actual inspection, different transmission structures have different transmission accuracy, and the appropriate transmission mechanism can be designed according to the above calculation results $\Delta w$.

Based on the above solution and in conjunction with the GCBDM imaging model designed in this article, it is possible to effectively address the bottlenecks in imaging detection schemes caused by inappropriate working distances, which lead to obstruction of the imaging optical path. It also resolves issues such as obstruction of the imaging optical path due to unsuitable lens focal lengths during the imaging process, or the inability to meet precision requirements for detection, which is shown in Fig 7.

Further, using the GCBDM established in this paper, the optimized detection model is automatically obtained by modifying the corresponding geometry parameters in different application scenarios, thus adapting to the detection needs of different samples to be detected. In this paper, we try different detection scenarios, taking different pharmaceutical packaging products as examples, such as bottom detection for cough syrup bottle and capsule medicine bottle, etc.

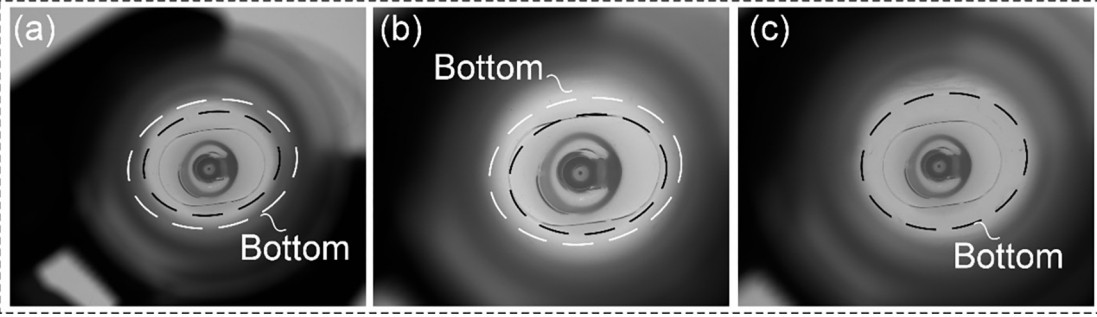

**Fig 7. Imaging optical path occlusion.** (a) Inappropriate working distance causes the imaging optical path to be obscured by the bottleneck. (b) Inappropriate lens focal length causes the imaging optical path to be obscured by the bottleneck. (c) Full imaging is possible, but due to inappropriate focal length or working distance, the accuracy cannot reach 0.05$mm$.

**Fig 8. Constraints of cough syrup bottle bottom detection and calculation process of detection model.**

## 4.2 Experiments of GCBDM on the bottle bottom of cough syrup

**4.2.1 Constraints of detection, and calculation of GCBDM.** For the bottle of cough syrup, its shape can be approximated as a cylinder with a tall bottle and a small neck. During the inspection, the information from the bottom needs to be imaged on the camera imaging plane through the narrow bottle neck. The parameter calculation process based on constraints $\beta < \delta, \beta = p/M$, $P/M > D_1$, $d > u > H$, and $w' < w$ is shown in Fig 8.

For the bottom detection of the cough syrup bottle, the main detection is the presence of foreign matter (e.g., hair, fiber, etc.), and the detection accuracy is about $0.1\,mm$. Likewise, the bottle has a narrow opening, and the width at the bottle neck position should not exceed the width of the narrow bottle neck.

According to the calculation of the detection model shown in Fig 8, and the commonly used imaging lens and camera parameters (which can be initially selected based on Eq (18)), the design of the detection model is shown in Table 4.

Based on the above calculations, the economical imaging schemes are selected as shown in Table 5. Since the 1.3MP sensor size and the 2.3MP sensor size can already meet the actual detection requirements, the use of the 2.3MP sensor with higher total number of pixels, resulting in increased costs and increased time for subsequent feature calculations.

**4.2.2 Experiment results and discussion.** According to the above two options, the detection models are built separately to clearly image the bottom through the narrow opening, and the results are shown in Fig 9.

As the experimental results shown in Fig 9, the different configurations of the above two options can achieve imaging of the bottom from above the bottle opening and clearly image

**Table 4. Magnification calculated based on constraints for different hardware conditions in the bottom detection of cough syrup bottle.**

|  |  | $M$ |
| --- | --- | --- |
| $f$ | $6mm$ | $0.0333 < M < 0.0448$ |
|  | $8mm$ | $0.0449 < M < 0.0606$ |
|  | $12mm$ | $0.0690 < M < 0.0938$ |
|  | $16mm$ | $0.0941 < M < 0.1290$ |
| $Pixel_{short}$ | 1024 | $0.0480 < M < 0.0793$ |
| $P$ | $4.8um$ |  |
| $Pixel_{short}$ | 1200 | $0.0480 < M < 0.0929$ |
| $p$ | $4.8um$ |  |
| $Pixel_{short}$ | 1536 | $0.0345 < M < 0.0855$ |
| $p$ | $3.45um$ |  |
| $Pixel_{short}$ | 2048 | $0.0345 < M < 0.1140$ |
| $p$ | $3.45um$ |  |

**Table 5. Feasible imaging solutions for the bottom detection of empty cough syrup bottle.**

| Option 1 | Camera: 2380×1024,4.8um×4.8um Lens: $f$ = 8mm/12mm $f$ = 8mm/12mm support sensor size 1/2″ | Approximate working distance: $f = 8mm$ : $140.013mm < u < 174.667mm$ $f = 12mm$ : $163.324mm < u < 185.913mm$ | $\Delta w = w-w'$ $f = 8mm$ : $2.595mm < \Delta w < 13.137mm$ $f = 12mm$ : $0.018mm < \Delta w < 5.553mm$ |
|---|---|---|---|
| Option 2 | Camera: 1920×1200,4.8um×4.8um Lens: $f$ = 8mm/12mm support sensor size 2/3″ | Approximate working distance: $f = 8mm$ : $140.013mm < u < 174.667mm$ $f = 12mm$ : $141.171mm < u < 185.913mm$ | $\Delta w = w-w'$ $f = 8mm$ : $2.595mm < \Delta w < 13.137mm$ $f = 12mm$ : $0.018mm < \Delta w < 12.702mm$ |

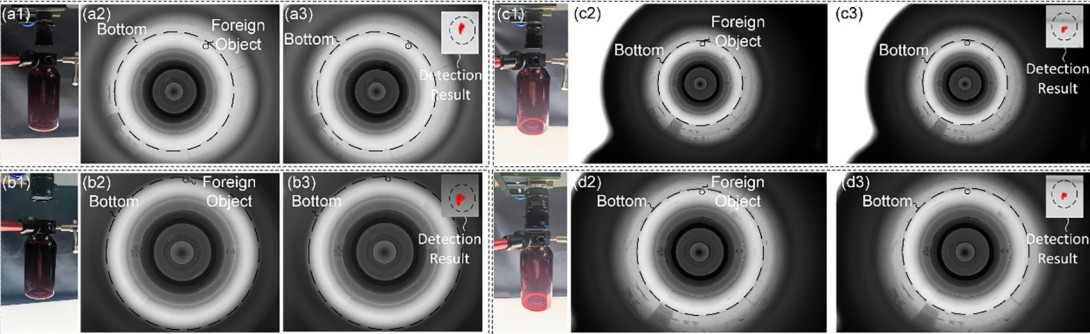

**Fig 9. Imaging results obtained according to different detection schemes.** (a1) (a2) (b1) (b2) the constructed detection models and the images of the bottom acquired by the 1.3MP camera at the lens focal length condition f = 8mm,12mm; (c1) (c2) (d1) (d2) the constructed detection models and the images of the bottom acquired by the 2.3MP camera at the lens focal length condition f = 8mm,12mm; (b3) (c3) (d3) the detection results of the traditional image processing with the Eqs (15) (16) and (17).

the visible foreign object present at the bottom. Besides, the maximum deviation $\Delta w$ of the center of the imaging from the center of the bottom is calculated under the conditions of the inspection model satisfying the geometric constraints. It can be seen from this that, for the method in option 1 of $f = 12mm$, the deviation range between the center of the photo and the center of the bottom is smaller, i.e., when the transmission mechanism produces a large deviation, the bottom information is more easily obscured by the mouth of the bottle. It can also be seen in Fig 9(b2) that the imaging information at the bottom is close to the imaging information at the opening, so that a slight transmission deviation can cause the information at the bottom to be interfered with by the mouth of the bottle. Additionally, in the process of inspecting cough syrup bottles, similar challenges arise due to bottlenecks caused by issues like working distance, lens focal length, and lens target surface, leading to obstructions in the imaging optical path and resulting in problems such as inability to achieve detection accuracy, decreased resolution, and distortion, as illustrated in Fig 10. Employing this solution can still effectively prevent these issues from occurring. Therefore, under the condition of ensuring the height of mechanical installation, the detection model of (a2) (c2) (d2) is preferred. On the one hand, it is possible to meet the test specifications, and on the other hand, it is possible to allow the drive mechanism to have a certain amount of motion error.

## 4.3 Experiment of GCBDM on the bottle bottom of capsule medicine

### 4.3.1 Constraints of detection, and calculation of GCBDM.
Further, for the bottom detection of common capsule medicine bottle, its shape can approximate a cylinder with a

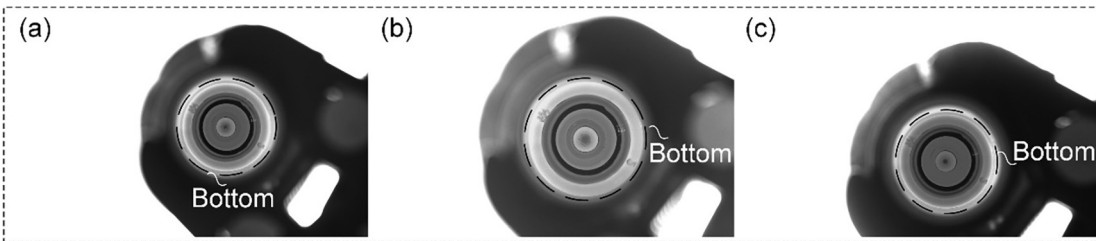

**Fig 10. Insufficient imaging accuracy.** (a) Inappropriate working distance results in the inability to achieve a detection accuracy of 0.05$mm$. (b) Inappropriate focal length results in the inability to achieve a detection accuracy of 0.05$mm$. (c) Due to factors such as working distance or focal length, as well as the imaging chip of the lens, the precision cannot reach 0.05$mm$.

$$\begin{cases} w = 38 \\ h = 17.5 \\ H = 107 \\ D_1 = D_2 = 62 \\ \delta = 0.1 \end{cases} \rightarrow \begin{cases} p/M < 0.1 \\ P/M > 62 \\ u > 107 \\ w' < 38 \end{cases} \rightarrow \begin{cases} \dfrac{p}{0.1} < M < \dfrac{P}{62} \\ \dfrac{1}{\dfrac{62 \cdot (107-17.5)}{f \cdot (62-38)} - 1} < M < \dfrac{1}{\dfrac{107}{f} - 1} \end{cases}$$

**Fig 11. Constraints of capsule medicine bottle bottom detection and calculation process of detection model.**

larger opening and broader imaging constraints compared to the two pharmaceutical packages mentioned above. According to the detection needs, the foreign matters (such as fiber hair, etc.) present at the bottom need to be detected with an accuracy of about 0.1$mm$. The parameter calculation process based on constraints $\beta < \delta, \beta = p/m$, $P/M > D_1$, and $w' < w$ is shown in Fig 11.

Based on the calculation process of the detection model shown in Fig 11, and the commonly used imaging lens and camera parameters (the initial selection is made according to Eq (18)), the design of the bottle bottom detection model for capsule medicine bottle is shown in Table 6.

Based on the above calculation results, the economical imaging schemes are selected as shown in Table 7.

**4.3.2 Experiment results and discussion.** According to the different configurations in the above two options, the detection models are built to clearly image the bottom through the opening of the bottle, and the results are shown in Fig 12.

As the experimental results shown in Fig 12, different configurations of the above two options can achieve clear imaging of the bottle bottom. Similarly, the maximum deviation range $\Delta w$ of the center position of the imaging from the center position of the bottle bottom is calculated under the condition of the detection model satisfying the geometric constraints.

As can be seen, for the configuration in option 1 of $f = 16mm$ and option 2 of $f = 16mm$, although the imaging can be satisfied with a high working distance, but when the drive mechanism has a large deviation, the imaging information at the bottom is more likely to be obscured by the bottle mouth. And from Fig 12 (d2) (h2), it can be seen that the imaging information of

**Table 6. Magnification calculated based on constraints for different hardware conditions in the bottom detection of capsule medicine bottle.**

| | | $M$ |
|---|---|---|
| $f$ | 6$mm$ | 0.0266<$M$<0.0594 |
| | 8$mm$ | 0.0358<$M$<0.0808 |
| | 12$mm$ | 0.0548<$M$<0.1263 |
| | 16$mm$ | 0.0744<$M$<0.1758 |
| $Pixel_{short}$ | 1024 | 0.0480<$M$<0.0793 |
| $P$ | 4.8$um$ | |
| $Pixel_{short}$ | 1200 | 0.0480<$M$<0.0929 |
| $p$ | 4.8$um$ | |
| $Pixel_{short}$ | 1536 | 0.0345<$M$<0.0855 |
| $p$ | 3.45$um$ | |
| $Pixel_{short}$ | 2048 | 0.0345<$M$<0.1140 |
| $p$ | 3.45$um$ | |

**Table 7. Feasible imaging solutions for the bottom detection of empty capsule medicine bottle.**

| Option 1 | Camera: 1280×1024,4.8um×4.8um Lens:$f = 6mm/8mm/12mm/16mm$ support sensor size 1/2″ | Approximate working distance: | $\Delta w = w - w'$ |
|---|---|---|---|
| | | $f = 6mm$ : $107.010mm < u < 131.000mm$ | $f = 6mm$ : $18.358mm < \Delta w < 27.855mm$ |
| | | $f = 8mm$ : $108.883mm < u < 174.667mm$ | $f = 8mm$ : $7.769mm < \Delta w < 26.963mm$ |
| | | $f = 12mm$ : $163.324mm < u < 230.978mm$ | $f = 12mm$ : $0.023mm < \Delta w < 9.975mm$ |
| | | $f = 16mm$ : $217.765mm < u < 231.054mm$ | $f = 16mm$ : $0.016mm < \Delta w < 1.481mm$ |
| Option 2 | Camera: 1920×1200,4.8um×4.8um Lens:$f = 6mm/8mm/12mm/16mm$ support sensor size 2/3″ | Approximate working distance: | $\Delta w = w - w'$ |
| | | $f = 6mm$ : $107.010mm < u < 131.000mm$ | $f = 6mm$ : $18.358mm < \Delta w < 27.855mm$ |
| | | $f = 8mm$ : $107.010mm < u < 174.667mm$ | $f = 8mm$ : $18.358mm < \Delta w < 26.963mm$ |
| | | $f = 12mm$ : $141.171mm < u < 230.978mm$ | $f = 12mm$ : $0.023mm < \Delta w < 15.306mm$ |
| | | $f = 16mm$ : $188.228mm < u < 231.054mm$ | $f = 16mm$ : $0.016mm < \Delta w < 5.480mm$ |

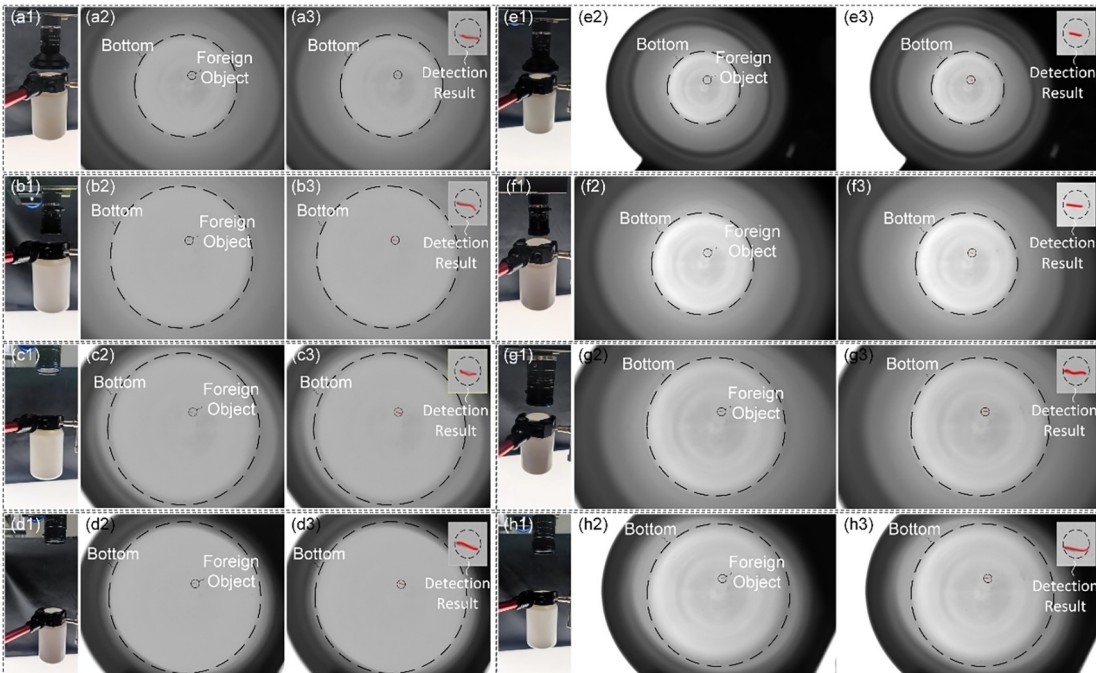

**Fig 12. Imaging results obtained according to different detection schemes.** (a1) (a2) (b1) (b2) (c1) (c2) (d1) (d2) the constructed detection models and the images of the bottom acquired by the 1.3MP camera at the lens focal length condition f = 6mm,8mm,12mm,16mm; (e1) (e2) (f1) (f2) (g1) (g2) (h1) (h2) the constructed detection models and the images of the bottom acquired by the 2.3MP camera at the lens focal length condition f = 6mm,8mm,12mm,16mm;(a3) (b3) (c3) (d3) (e3) (f3) (g3) (h3) the detection results of the traditional image processing with the Eqs (15) (16) and (17).

the bottom is close to the imaging information of the bottle mouth, i.e., the center of the photographed picture is slightly deviated from the center of the bottom of the bottle, which will make the information of the bottom interfered by the bottle opening in the imaging optical path. In addition, for the inspection of the capsule medicine bottle in Fig 12 (a2) (b2) (e2) (f2), due to the short focal length of the lens with a large divergence angle, it is possible to simultaneously image the information of the bottom and the information of the inner wall of the bottle on the camera imaging plane during the inspection, thus enabling the detection of the inner wall of the bottle. Moreover, due to the capsule bottles having wider mouths and shorter bodies, the application of the solution proposed in this article can more effectively avoid issues related to working distance, lens focal length, and lens target surface that are common in traditional imaging models, as shown in Fig 13. This allows for more effective avoidance of these issues and better imaging detection.

Similarly, it can be seen from the results that the height of the test should be appropriately reduced under the condition that the height of the mechanical installation is allowed to ensure sufficient imaging size of the bottom of the bottle. On the one hand, it ensures sufficient imaging size of the bottle bottom, and on the other hand, it can give a certain motion error to the drive mechanism.

## 5 Conclusion

This paper takes the bottom detection of plastic LVP empty bottle as the research object. Its bottle shape has irregular characteristics, as well as the characteristic of the narrow bottle opening, and its detection accuracy is much higher than the traditional bottle bottom detection requirements. The existing inspection methods cannot establish a universal visual inspection

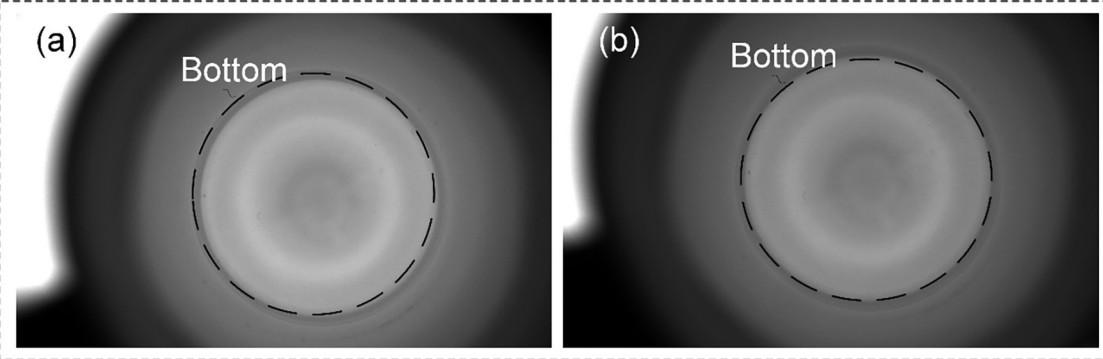

**Fig 13. Imaging situation for an unsuitable plan.** (a) Inappropriate working distance results in the inability to achieve a detection accuracy of 0.05$mm$. (b) Due to factors such as working distance or focal length, as well as the imaging chip of the lens, the precision cannot reach 0.05$mm$.

model based on its inspection accuracy, the shape of the inspected bottle and the inspection field of view. In order to ensure the effectiveness of the imaging optical path, this paper proposes a geometric constraint-based detection model (GCBDM), according to the requirements of the detection accuracy (imaging accuracy) and detection field of view, combined with the imaging model and the shape characteristics of the bottle to be detected, to complete the imaging model design and optimization of the bottle bottom. Moreover, based on the above generic detection model, it can be adapted to different detection needs by easily modifying the target parameters for different application scenarios. In addition, by calculating the maximum deviation range of the imaging center position and the center position of the bottle bottom to be detected, it can provide a basis for the design of the accuracy of the transmission structure in the detection, to ensure that in the motion detection, the error of the transmission will not cause the bottle bottom information in the imaging optical path to be obscured by the narrow bottle opening, so as to ensure the stability of the detection. Moreover, the geometric constraint-based detection model (GCBDM) proposed in this paper not only surpasses traditional bottle bottom detection requirements in terms of detection accuracy but also exhibits significant improvements in detection speed, stability, and adaptability to detection environments compared to traditional detection methods. With only appropriate adjustments to the system's parameters, the GCBDM detection model designed in this article can be fully applicable to different industrial scenarios. For example, some large domestic intravenous infusion manufacturers have already applied the model proposed in this article to their actual production processes. However, some smaller-scale enterprises might still opt for manual inspection due to cost considerations. The GCBD detection model designed in this paper, with minor parameter adjustments, can not only be applied in the medical industry for detecting the bottom of bottles but also has great potential for widespread application in fields like the bottom detection of liquor bottles and perfume bottles.

## Supporting information

**S1 File. Principle of image preprocessing algorithm.**
(DOCX)

## Author Contributions

**Conceptualization:** Pi Yuan, Chen Li, Yongjing Yin.

**Data curation:** Pi Yuan, Peng Tang, Yongjing Yin.

**Formal analysis:** Pi Yuan, Peng Tang, Yongjing Yin.

**Investigation:** Pi Yuan.

**Methodology:** Pi Yuan, Chen Li.

**Project administration:** Peng Tang, Bin Yuan, Yongjing Yin.

**Software:** Pi Yuan.

**Supervision:** Bin Yuan, Yongjing Yin.

**Validation:** Pi Yuan, Chen Li.

**Writing – original draft:** Pi Yuan.

**Writing – review & editing:** Chen Li, Peng Tang, Bin Yuan.

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
