## [Decision Letter · Decision Letter 0]

7 Nov 2023

PONE-D-23-14713

Machine vision model for detection of foreign substances at the bottom of empty large volume parenteral

PLOS ONE

Dear Dr. Li,

Thank you for submitting your manuscript to PLOS ONE. After careful consideration, we feel that it has merit but does not fully meet PLOS ONE’s publication criteria as it currently stands. Therefore, we invite you to submit a revised version of the manuscript that addresses the points raised during the review process.

Please, address all the comments made by the reviewers. 

We look forward to receiving your revised manuscript.

Kind regards,

Antonio Riveiro Rodríguez, PhD

Academic Editor

PLOS ONE

Journal Requirements:

"This work was supported by Zhejiang Provincial Natural Science Foundation of China under Grant No. LGF20F050002，National Natural Science Foundation of China under Grant No. 62103340, and Hangzhou City Agriculture and Social Development General Project under Grant No. 20191203B34 and No.20201203B118."             

Reviewers' comments:

Reviewer's Responses to Questions

**Comments to the Author**

1. Is the manuscript technically sound, and do the data support the conclusions?

Reviewer #1: Partly

Reviewer #2: No

Reviewer #3: Yes

2. Has the statistical analysis been performed appropriately and rigorously? 

Reviewer #1: N/A

Reviewer #2: I Don't Know

Reviewer #3: Yes

3. Have the authors made all data underlying the findings in their manuscript fully available?

Reviewer #1: No

Reviewer #2: Yes

Reviewer #3: Yes

4. Is the manuscript presented in an intelligible fashion and written in standard English?

Reviewer #1: Yes

Reviewer #2: No

Reviewer #3: Yes

5. Review Comments to the Author

Reviewer #1: Machine vision model for detection of foreign substances at the bottom of empty large volume parenteral

Recommendation: Major Revision

Comments: The paper motivation is not state-of-art, however there are some major revisions and some minor concerns that needs to be addressed.

Major Revision:

1. Cite the traditional method for the detection of LVP in the introduction.

2. The conditions which are mentioned in section 2 are not properly explained. What are the criteria of the selections of these conditions?

3. The authors used the images for equation in the explanation of fig. 5, explain it properly otherwise cite it.

4. The contribution of this work is not well defined and needs to be properly defined.

5. Motivation of the works is not well-defined, what is the need of proposing this new method?

6. Compare the proposed methods results with existing methods.

Minor Concerns:

1. Display the high-quality image of LVP bottle, which show the foreign substance.

2. Any reference of pin-hole imaging model?

3. What is the principle of lens imaging? And its benefits and limitations?

4. Similar triangles reference?

5. Write the equations in standard format.

6. Equation 7 needs more explanation mathematically.

7. Parameters of common industrial cameras and lenses, any benchmark reference?

7. How equation 15 is generated, explain it.

8. Add the space between reference number and last word of sentence.

Reviewer #2: The implication from the title and abstract is that the authors are presenting a machine vision model for automated detection of foreign substances in empty LVPs.

However, the presented result is only about the detection of container bottoms. Manual detection of foreign substances is alluded to only as the motivating application for container bottom detection. The title and abstract are highly misleading as no method for foreign object detection is presented.

Furthermore, even in the limited application of focusing on container bottoms, I am not convinced by the article of this being an unsolved problem. I am skeptical that standard auto-focus approaches could not easily image the bottom, or that focal sweeps with extended depth-of-field compositing could not be used.

If the authors want to focus on foreign object detection for this paper, I would encourage them to add some automated image analysis. Under the highly controlled lighting conditions of the experiment setup, creating an image classifier that separates "clean" from "contaminated" should be reasonably straightforward.

Reviewer #3: Summary:

The paper addresses the challenge of bottom detection in plastic LVP empty bottles, which possess irregular shapes and narrow openings. The authors propose a Geometric Constraint-Based Detection Model (GCBDM) to enhance the accuracy and adaptability of the detection process. By integrating the imaging model with the bottle's shape characteristics, the proposed method claims to be superior to traditional inspection methods. The paper also delves into the modification of target parameters to suit varied application scenarios, ensuring detection stability and accuracy.

Strengths:

Originality: The paper introduces a novel model (GCBDM) that fills a gap in the current literature and has the potential to revolutionize the bottom detection process for irregularly shaped bottles.

Adaptability: The model's ability to be adapted for different scenarios is a significant strength, catering to a variety of inspection needs.

Detailed Analysis: The authors provide a comprehensive analysis of the maximum deviation between the imaging center position and the bottle bottom's center, which ensures the reliability of the detection process.

Areas for Improvement and Recommendations:

Clarity in Methodology: While the GCBDM is introduced, the methodology section could be elaborated further. It might be beneficial to include step-by-step algorithms or flowcharts to illustrate the process.

Scalability: The authors should discuss the model's scalability. Can it be used on an industrial scale, and if so, what are the potential challenges or limitations?

Case Studies: The inclusion of real-world case studies where the model has been implemented would provide readers with practical insights and applications of the model.

Future Prospects: It would be intriguing to understand any potential extensions or improvements to GCBDM that the authors foresee in future works.

Conclusion:

The paper offers an exciting approach to a niche yet significant problem in bottle inspection. With further elaboration, empirical validation, and real-world testing, the proposed model can become a benchmark in the domain. I recommend the paper for publication post the suggested revisions.

6. PLOS authors have the option to publish the peer review history of their article (what does this mean?). If published, this will include your full peer review and any attached files.

Reviewer #1: No

Reviewer #2: No

Reviewer #3: No

---

## [Author Response · Author response to Decision Letter 0]

20 Dec 2023

Editor, Concern #1. Please ensure that your manuscript meets PLOS ONE's style requirements, including those for file naming. 

Author response: Thank you for your comments. 

Author action: We have modified the manuscript format as requested.

Editor, Concern #2. Please state what role the funders took in the study.

Author response: Thank you for your comments. 

Author action: The fund supported the corresponding author Chen Li. Author Chen Li participated in the writing of the paper and the verification and guidance of the experiments.

Editor, Concern #3. In your Data Availability statement, you have not specified where the minimal data set underlying the results described in your manuscript can be found.

Author response: Thank you for your comments. 

Author action: https://figshare.com/account/home

Email address: yp15237870669@163.com. Password: ISCI20211111@a

All the data on the figshare page includes all the files from the manuscript and supplementary information.

Reviewer#1, Concern # 1: Cite the traditional method for the detection of LVP in the introduction.

Author response: Thank you for your comments. 

Author action: We have updated the introduction of the manuscript.

Reviewer#1, Concern # 2: The conditions which are mentioned in section 2 are not properly explained. What are the criteria of the selections of these conditions?

Author response: Thank you for your comments.

Author action: We have added the criteria standards.

Reviewer#1, Concern # 3: The authors used the images for equation in the explanation of fig. 5, explain it properly otherwise cite it.

Author response: Thank you for your comments. The formula in Fig. 5 represents the actual dimensions of the object under inspection and the parameters of the industrial camera and lens, based on the calculation process of the constraints mentioned earlier. For the convenience of formatting, they were all written together in Visio software.

Reviewer#1, Concern # 4: The contribution of this work is not well defined and needs to be properly defined.

Author response: Thank you for your comments.

Author action: We have added contributions from different authors.

Reviewer#1, Concern # 5: Motivation of the works is not well-defined, what is the need of proposing this new method?

Author response: Thank you for your comments.

Author action: Currently, the detection in the field of large infusion is mainly manual, which cannot meet the production speed and production conditions of the production line. The research of this article can make up for this deficiency. Firstly, the method proposed in this article can improve the efficiency and automation of the production line, reduce labor costs, and error rates. Secondly, through automatic detection, defects such as empty bottles can be detected more quickly and accurately to ensure the quality and safety of infusion products. In addition, machine vision technology can realize 24/7 all-weather monitoring, improve the traceability and quality control level of the production process, which is of great significance for the medical device industry.

Reviewer#1, Concern # 6: Compare the proposed methods results with existing methods.

Author response: Thank you for your comments.

Author action: We have updated the manuscript by adding a discussion section.

Reviewer#1, Concern # 7: Display the high-quality image of LVP bottle, which show the foreign substance.

Author response: Thank you for your comments.

Author action: We have updated the images.

Reviewer#1, Concern # 8: Any reference of pin-hole imaging model?

Author response: Thank you for your comments.

Author action: We have updated the references.

Reviewer#1, Concern # 9: What is the principle of lens imaging? And its benefits and limitations?

Author response: Thank you for your comments. 

The industrial lens imaging principle of pinhole imaging relies on using a very small aperture or pinhole to restrict the entry of light into the lens and form an image. When light passes through the pinhole, diffraction occurs, causing the light to converge and form an inverted real image. This imaging principle can be employed in industrial lenses, such as in microscopes and other optical devices. Pinhole imaging can provide high depth and clarity as it reduces light scattering, resulting in clear images. However, pinhole imaging also has limitations, with the primary one being that the aperture's restriction of light reduces the amount of light entering the lens, leading to relatively dim images. Additionally, the presence of diffraction can cause some details in the image to appear blurry. Therefore, when considering the use of pinhole imaging, it is essential to weigh its advantages and limitations to ensure it aligns with the specific requirements of the application.

Reviewer#1, Concern # 10: Similar triangles reference?

Author response: Thank you for your comments. The translation of this sentence into English is: "This section does not require referencing literature, and the theoretical knowledge is based on the use of the similar triangle model in mathematical models.

Reviewer#1, Concern # 11: Write the equations in standard format.

Author response: Thank you for your comments.

Author action: We have updated the manuscript.

Reviewer#1, Concern # 12: Equation 7 needs more explanation mathematically.

Author response: Thank you for your comments.

Author action: We have updated the manuscript.

Reviewer#1, Concern # 13: Parameters of common industrial cameras and lenses, any benchmark reference?

Author response: Thank you for your comments.

The common parameters of industrial cameras and lenses include resolution, sensor size, focal length, aperture, field of view angle, etc. These parameters typically affect image quality, applicability, and cost. For industrial cameras, the common benchmark references are pixel size and sensor size. Pixel size is usually used to measure image quality, while sensor size affects the field of view angle and applicability. In addition, shutter speed, dynamic range, and signal-to-noise ratio are also important indicators for evaluating the performance of industrial cameras. As for industrial lenses, the common benchmark references are focal length, aperture, and field of view angle. Focal length determines the lens's magnification, aperture affects the amount of light and depth of field, and field of view angle determines the lens's shooting range. Furthermore, lens distortion, chromatic aberration, and lens surface quality are also important parameters for evaluating lens performance. To select suitable industrial cameras and lenses, key parameters can be determined based on specific application requirements, and reference can be made to the technical specifications and performance data provided by manufacturers. Additionally, actual testing and comparison can be conducted to verify their performance in specific scenarios, ensuring the selection of the most suitable equipment.

Reviewer#1, Concern # 14: How equation 15 is generated, explain it.

Author response: Thank you for your comments. Formula 15 is derived to calculate the minimum resolution required for a camera based on the minimum field of view and detection accuracy needed during the detection process. The camera's field of view divided by the per-pixel accuracy equals the camera resolution. Using this formula, one can calculate the necessary camera specifications for practical applications and select an appropriate camera.

Reviewer#1, Concern # 15: Add the space between reference number and last word of sentence.

Author response: Thank you for your comments.

Author action: We have updated the format.

Reviewer#2, Concern # 1: However, the presented result is only about the detection of container bottoms. Manual detection of foreign substances is alluded to only as the motivating application for container bottom detection. The title and abstract are highly misleading as no method for foreign object detection is presented.

Author response: Thank you for your comments.

Author action: We have updated the format.

Reviewer#2, Concern # 2: Furthermore, even in the limited application of focusing on container bottoms, I am not convinced by the article of this being an unsolved problem. I am skeptical that standard auto-focus approaches could not easily image the bottom, or that focal sweeps with extended depth-of-field compositing could not be used.

Author response: Thank you for your comments. If an autofocus system is used, the unique shape of the bottom of a large infusion bottle may prevent the autofocus from accurately determining the optimal focus position. Additionally, the autofocus system might exhibit errors or instability. In some cases, autofocus could be affected by environmental factors, lens quality, or camera hardware issues, leading to focus shifts or instability. This paper adopts a pinhole imaging model, where the information from the bottom of the bottle is projected through the narrow bottleneck onto the image sensor. By ensuring that the working distance of the imaging unit is greater than the height of the bottle, defocusing issues are avoided, ensuring clear imaging. Furthermore, the actual production process requires a rate of 18,000 pcs/h. If technologies such as autofocus and depth of field combination focus scanning are used, it would consume a significant amount of time and fail to meet the actual production needs.

Reviewer#2, Concern # 3: If the authors want to focus on foreign object detection for this paper, I would encourage them to add some automated image analysis. Under the highly controlled lighting conditions of the experiment setup, creating an image classifier that separates "clean" from "contaminated" should be reasonably straightforward.

Author response: Thank you for your comments.

Author action: We have updated the format.

Reviewer#3, Concern # 1: Clarity in Methodology: While the GCBDM is introduced, the methodology section could be elaborated further. It might be beneficial to include step-by-step algorithms or flowcharts to illustrate the process.

Author response: Thank you for your comments.

Author action: We have updated the format.

Reviewer#3, Concern # 2: Scalability: The authors should discuss the model's scalability. Can it be used on an industrial scale, and if so, what are the potential challenges or limitations?

Author response: Thank you for your comments.

Author action: We have updated the format.

Reviewer#3, Concern # 3: Case Studies: The inclusion of real-world case studies where the model has been implemented would provide readers with practical insights and applications of the model.

Author response: Thank you for your comments.

Author action: We have updated the format.

Reviewer#3, Concern # 4: Future Prospects: It would be intriguing to understand any potential extensions or improvements to GCBDM that the authors foresee in future works.

Author response: Thank you for your comments.

Author action: We have updated the format.

---

## [Decision Letter · Decision Letter 1]

20 Jan 2024

Machine vision model for detection of foreign substances at the bottom of empty large volume parenteral

PONE-D-23-14713R1

Dear Dr. Li,

We’re pleased to inform you that your manuscript has been judged scientifically suitable for publication and will be formally accepted for publication once it meets all outstanding technical requirements.

Kind regards,

Antonio Riveiro Rodríguez, PhD

Academic Editor

PLOS ONE

Reviewers' comments:

Reviewer's Responses to Questions

**Comments to the Author**

1. If the authors have adequately addressed your comments raised in a previous round of review and you feel that this manuscript is now acceptable for publication, you may indicate that here to bypass the “Comments to the Author” section, enter your conflict of interest statement in the “Confidential to Editor” section, and submit your "Accept" recommendation.

Reviewer #1: All comments have been addressed

2. Is the manuscript technically sound, and do the data support the conclusions?

Reviewer #1: Yes

3. Has the statistical analysis been performed appropriately and rigorously? 

Reviewer #1: I Don't Know

4. Have the authors made all data underlying the findings in their manuscript fully available?

Reviewer #1: Yes

5. Is the manuscript presented in an intelligible fashion and written in standard English?

Reviewer #1: Yes

6. Review Comments to the Author

Reviewer #1: (No Response)

7. PLOS authors have the option to publish the peer review history of their article (what does this mean?). If published, this will include your full peer review and any attached files.

Reviewer #1: No

---

## [Editor Report · Acceptance letter]

6 Mar 2024

PONE-D-23-14713R1 

PLOS ONE

Dear Dr. Li, 

I'm pleased to inform you that your manuscript has been deemed suitable for publication in PLOS ONE. Congratulations! Your manuscript is now being handed over to our production team.

Kind regards, 

on behalf of

Dr. Antonio Riveiro Rodríguez 

Academic Editor

PLOS ONE